# Phosphorus and carbon in soil particle size fractions – A global synthesis

Marie Spohn<sup>1</sup>


<sup>1</sup>Soil Biogeochemistry, Bayreuth Center of Ecology and Environmental Research (BayCEER), University of Bayreuth, Bayreuth, Germany

Correspondence to: Marie Spohn (marie.spohn@uni-bayreuth.de)

Abstract. Despite the importance of phosphorus (P) as a macronutrient, the factors controlling storage of organic
 phosphorus (OP) in soils are not yet well understood. The objective of this meta-analysis was therefore to investigate the distribution of OP, organic carbon (OC), and inorganic P (IP) in particle size fractions depending on climate, latitude and land use, based on data from published studies. The clay size fraction contained on average
 8.8 times more OP than the sand size fraction and 3.9 and 3.2 times more IP and OC, respectively. The OP concentrations of the silt size and clay size fractions were both most strongly correlated with mean annual

- temperature (MAT) ( $R^2=0.30$  and 0.31, respectively, p<0.001). Latitude, MAT and mean annual precipitation together largely explained the variability of the OC concentration of the clay size fraction ( $R^2=0.73$ , p<0.001). The OC:OP ratios of the silt and clay size fraction were correlated with latitude ( $R^2=0.49$  and 0.34, respectively, p<0.001), and the OC:OP ratio of the clay size fraction changed less strongly with latitude than the OC:OP ratio of the silt and the sand size fraction. The OC concentrations of all particle size fractions were significantly (p<0.05)
- 20 lower in the croplands than in the adjacent soils under (semi-)natural vegetation. In contrast, the OP concentration was only significantly (p<0.05) decreased in the sand size fraction due to land use conversion. In conclusion, this meta-analysis shows that OP concentrations in the silt and clay size fraction strongly depend on climate and latitude, and that OP is more strongly enriched in the clay size fraction than OC and IP, which is likely due to the fact that OP competes very successfully for sorption sites in soil. The strong sorption of OP in soil, especially in</p>
- the clay size fraction, makes OP less vulnerable to land-use change than soil OC.

## 1. Introduction

Phosphorus (P) is a key macronutrient necessary to all living organisms. During the next decades, P supply for agricultural production will likely shorten (Cordell et al., 2009). Therefore, it is important to better understand storage and sorption of organic P (OP) and inorganic P (IP) in soils (Turner et al., 2005; Georg et al., 2018).

- Organisms use P to form tissues (apatite in bones and teeth, phospholipids in cell membranes), carry genetic information (deoxyribonucleic acid (DNA), ribonucleic acid (RNA)), and store energy (adenosine triphosphate (ATP)). In soils, inorganic P (IP) is mostly comprised of orthophosphate and to a much lesser extent of polyphosphates. In contrast, OP occurs in many different forms, and the dominant OP compounds are, first, phosphomonoesters (PME) that comprise inositol phosphatase and other, more labile PME (such as ATP), and
- second, phosphodiesters (PDE; such as DNA and RNA). Overall, inositol phosphates tend to accumulate in the environment, while PDE and labile PME are less prevalent (Steward and Tiessen, 1987; Darch et al., 2014).

The protonation of all phosphate groups depends on soil pH. In moderately acid to acid soils, phosphates are mostly present as monovalent anions. Only in alkaline soils, phosphates are to a larger extent present in the form of bivalent anions. Being anions, phosphates sorb to positively charged surfaces in soil such as Fe and Al oxides and

hydroxides as well as positively charged binding sites on organic matter (OM) and at the edges of phyllosilicates (Hinsinger, 2001). In addition, phosphates can also sorb to negatively charged surfaces through polyvalent metal cations that are attracted by the two negative charges (Kleber et la., 2007).

Due to its high charge density, resulting from multiple phosphate groups, inositol-hexa-phosphate as well as inositol-penta-, -tetra-, and -tri-phosphate has a higher capacity to compete for binding sites in soil than other

- PMEs and orthophosphate (McKercher and Anderson, 1989; Martin et al., 2004; Celi and Barberis, 2005; Ruttenberg and Sulak, 2011). In contrast, PDEs have a lower charge density than orthophosphate and PME and their phosphate groups are considerably shielded from ionic interactions. Thus, PDE have a lower capacity to compete for sorption sites in soils than orthophosphate and PME (Tate, 1984; Steward and Tiessen, 1987; Darch et al., 2014), and their capacity to successfully compete for sorption sites decreases with increasing molecular
- weight (Ogram et al., 1994). Most non-phosphorylated organic compounds, seem to have a lower capacity to compete for binding sites than OP compounds (Guppy et al., 2004; Fransson and Jones, 2007), and the addition of inorganic P to soil can prevent sorption of dissolved OC to the soil solid phase (Schneider et al., 2010).

Due to the high capacity of OP to sorb to mineral surfaces in soil, OP plays very likely an important role for the formation of organo-mineral complexes. However, this topic has received very little attention so far, despite the fort that are a mineral complexes are important and to play a mineral topic despite the formation of OM.

fact that organo-mineral complexes are increasingly esteemed to play an important role in the stabilization of OM against microbial decomposition in soil (von Lützow et al, 2006; Kleber et al., 2007; Kögel-Knabner et al., 2008; Schmidt et al., 2013).

Organo-mineral complexes have been intensively studied using particle size fractionation, which consists of mechanical destruction and dispersion of the soil sample followed by separation of particle size fractions through sieving and gravitational separation. Particle size fractionation is based on the concept that OM associated with

sieving and gravitational separation. Particle size fractionation is based on the concept that OM associated with particles of different size and therefore also of different mineralogical composition differs in structure and function (Christensen, 2001). While quartz particles that dominate the sand size fraction exhibit only weak bonding

affinities to OM, clay size particles (such as sesquioxides and phyllosilicates) provide a large surface area and numerous reactive sites, where OM can sorb (Sposito et al., 1999; Christensen, 2001; von Lützow et al., 2006).

Assuming that sorption is an important stabilization mechanism, OM in the sand size fraction is considered the active pool, OM in the silt size fraction the intermediate pool, and OM in the clay size fraction is considered the

- passive pool (von Lützow et al., 2006). Evidence for this comes from the observation that OM in the clay size fraction is older and has a longer turnover time than OM in the sand and in the silt size fraction (Anderson and Paul, 1984; Scharpenseel and Becker-Heidmann, 1989; Balesdent, 1996; Quideau et al., 2001; Eusterhues et al., 2003; Ludwig et al., 2003; Bol et al., 2009). Particle size fractionation has been used in many studies to gain insight into the effects of land use and depth distributions on OC (Christensen, 2001; von Lützow et al., 2007), and to a
- lesser extent of OP (von Sperber et al., 2017).

While sorption stabilizes OM against microbial decomposition, and thus affects mineralization of OP and OC and the soil's output of C in the form of CO<sub>2</sub>, the soil OC-to-OP ratio might also be affected by the inputs, i.e., by plant litter. The concentration of P in plant leaf litter depends on climate, latitude and biome. For forest biomes, it has been shown in a meta-analysis that the molar C:P ratio of leaf litter decreases in the order tropical forest < temperate

- coniferous forest < temperate broadleaf forest, and reaches a global mean of 1334 (McGroddy et al., 2004). In a second large meta-analysis on senesced plant litter in different biomes, it was found that the average molar C:P ratio across different plant functional types amounted to 1183, and increased with mean annual temperature (MAT) and mean annual precipitation (MAP), whereas the P concentrations exhibited the opposite trend (Yuan and Chen, 2009). While the global patterns of P in plant biomass and plant litter have been studied quite intensively</p>
- (McGroddy et al., 2004; Reich and Oleksyn, 2004; Yuan and Chen, 2009), much less is known about the global distribution of OC and OP in soils (Kirkby et al., 2011)

The objective of this study was to analyze the distribution of OP, OC, and IP across particle size fractions depending on geographical location of the soils, soil depth and land use based on data from published studies, with the aim to gain insights into the sequestration OP in soils and soil particle size fractions. I tested the hypotheses

that (i) OP is more strongly enriched in the clay size fraction with respect to the sand size fraction than OC and IP, (ii) the OP concentration in the clay size fraction is less affected by latitude, climate and land use than the OP concentration of the sand and silt size fraction, and (iii) that OP concentration of all particle size fractions is less affected by latitude, climate and land use than OC because of the stronger sorption of OP than of OC to soil minerals.

## 30

# 2. Material and Methods

#### 2.1 Dataset

I searched for peer-reviewed studies that report OP concentrations in particle size fractions of soils. All particle size fractions had to be gained by mechanical destruction and dispersion of the soil sample followed by separation

of the particle size fractions through sieving and gravitational separation, in order to be considered in the metaanalysis. Studies reporting OP contents in water-stable aggregates, in density fractions, etc. were excluded. All studies had to report OP concentrations determined either according to Saunders and Williams (1955) or as the sum of at least two organic Hedley fractions (Hedley, 1982) in order to be included in the meta-analysis. The IP

concentration in turn, had to be calculated either as the difference between total P and organic P following Saunders and Williams (1955) or as the sum of all inorganic Hedley fraction, respectively, according to Hedley (1982). Studies reporting OP in less than two Hedley fractions were excluded. All studies had to report OP concentrations of the three particle size fractions, namely sand, silt, and clay. Studies that reported OP concentrations of only one

5 or two particle size fractions were not included in the meta-analysis. If studies reported OP concentrations for more than three fractions (for example, separately for coarse silt and fine silt or coarse sand and fine sand), a weighted mean based on the masses of the two sand or two silt size fractions, respectively, was calculated. Only in a few cases, in which the masses of the fractions were not reported, means were calculated.

Besides OP concentrations in the particle size fractions, the following variables were extracted from the studies;
latitude, MAP and MAT of the study site, the country where the study site was located, the soil order, the land use type, the name and the depth of the soil horizon. If the latitude was not reported, it was retrieved from digital maps based on site name, or other descriptors. Furthermore, the following soil chemical variables were extracted; total organic carbon (TOC), total inorganic phosphorus (TIP), and TOP of the unfractionated soil as well as organic carbon (OC) and inorganic phosphorus (IP) concentrations of each particle size fraction. Moreover, the ratio of

- phosphomonoesters-to phosphodiesters (PME:PDE ratio) determined by nuclear magnetic resonance (NMR) spectroscopy, was collected from the studies as well as the mass of each particle size separate (in percentage of total). In case data were reported in graphs, data were acquired directly from the authors or were extracted from the graphs using the open-source software DataThief (Tummers, 2006). For the sake of clarity, the terms TOC, TOP, and TIP will be used only when referring to the bulk soil, and OC, OP and IP when referring to the
- concentrations of OC, OP and IP in the particle size fractions in the following. It should be noted that in the entire meta-analysis element concentrations of bulk soil and particle size fractions are considered but not element stocks.

In total, I found 11 peer-reviewed studies (reported in 13 publications, see Supplement 1) that met the criteria of the literature search (Table 1). The studies reported data on the OP concentrations in particle size fractions in 118 soil horizons located in 12 different countries at latitudes ranging from  $3^{\circ}$  to  $57^{\circ}$  (Table 1, see Supplement 2). The

25 total number of topsoil horizons amounted to 80, and for 16 of them, data on the PME:PDE ratio was reported (Table 1). Of the 118 soil horizons, 43 horizons were part of soil profiles, for which data on three or more horizons was provided. In addition, 10 land use type comparisons for topsoils were found, each consisting of a soil at a (semi-)natural site and a soil at an adjacent cropland site with comparable soil properties (Table 1).

#### 30 2.2 Data analysis

The data were harmonized (i.e. units were converted, if necessary). Molar ratios of TOC:TOP, TIP:TOP of the bulk soils were calculated as well as molar OC:OP and IP:OP ratios of the particle size fractions. The concentrations of OP, OC IP, and P as well as the OC:OP and IP:OP ratios across all three particle size fractions were compared using ANOVA (see below). Furthermore, I calculated linear (multiple) regression models for total

35 element concentrations and element ratios in the particle size fractions of the topsoils as a function of latitude, MAT and MAP. Subsoil horizons were not included in these analyses in order to avoid autocorrelation and dependence of data. For all analyses including latitude, only the degree of latitude was considered, but no differentiation between Southern and Northern hemisphere was made.

In order to learn about the quality of OP in the particle size fractions in the topsoils, the NMR data on OP species in the particle size fractions were analyzed. For this purpose, the PME:PDE ratios of all three particle size fractions were compared using ANOVA (see below). This comparison was conducted separately for soils of the tropics and the temperate zone. Studies that did not report the PME:PDE ratio for all three particle size fractions (sand, silt and clay size) were not included in the analysis.

In order to analyze the effect of land use conversion from natural or semi-natural vegetation to cropland on the distribution of OP and OC in the particle size fractions, a meta-analysis on all data on land use comparison was conducted. For this purpose, only studies that reported the OP concentration of the particle size fractions of two comparable soils from the same area but under different forms of land use were considered. If a study compared a

- native site and several arable sites, only the arable site with the longest duration of arable land use was considered. The TOP, TOC and TIP concentrations, and OP, OC, and IP concentrations of the particle size fractions in the topsoil were calculated. The TOP, TOC and TIP concentrations, and OP, OC, and IP concentrations of the particle size fractions of the native sites and the cropland sites were compared using ANOVA (see below). The change in TOP, TOC, TIP and TP as well as OP, OC, and IP concentrations in the particle size fractions in the topsoil due
- to land use conversion from native or semi-native vegetation to cropland was calculated separately for each land use comparison. Based on the analyses of the single land use comparisons, means and standard deviations across all 10 comparisons were calculated. The changes in TOP, TOC, TOP and TP as well as in OP, OC, and IP concentrations of the particle size fractions were compared using ANOVA (see below).
- In order to learn about the depth distribution of OP and IP in particle size fractions, an analysis of the profile data 20 was conducted. For this analysis, I considered only data on soil horizons that were reported together with data on at least two or more soil horizons from the same soil. The mean depth of all soil horizons was calculated, and monoexponential models of the OP concentrations of the particle size fractions as a function of soil depth were fitted to the data. In addition, linear regression models of the IP:OP ratios of each of the three particle size fractions as a function of soil depth were fitted to the data.
- Before calculating the linear regressions of TOC and TOP and of OC and OP in each of the three particle size fractions, the data were log transformed in order to achieve normal distribution. For all other regression analyses, no data transformation was required. Differences between particle size fractions and between different variables were tested by ANOVA followed by Tukey post-hoc test. In all analyses  $\alpha$ =0.05 was considered the threshold for significance. All analyses were conducted in R (R Core Team, 2013).

# 3. Results

#### 3.1 Distribution of C and P in particle size fractions in topsoils

In the topsoils, the mean concentrations of OP, IP, OC and P (sum of OP and IP) in the clay size fraction were significantly (p<0.001) higher than in the sand size fraction (Fig. 1a-d). The clay size fraction contained on average</li>
8.8 times more OP than the sand size fraction and 3.9, 3.2 and 5.1 times more IP, OC, and P, respectively (Fig. 1a-d). Thus, the ratio of clay size fraction-to-sand size fraction was 2.8 times larger for OP than for OC, and 2.3 times larger for OP than for IP. As a result of the unequal distribution of OC and OP across the three particle size fractions, the OC:OP ratio of the clay size fraction was significantly (p<0.05) lower than the one of the sand size</li>

fraction, on average by a factor of 3.8 (Fig. 1e). The molar OC:OP ratios amounted on average to 771, 424, and 204, in the sand, silt and clay size fraction, respectively. Due to the unequal distribution of IP and OP, the IP:OP ratio was significantly higher (p<0.001) in the sand size fraction than in the clay size fraction, on average by a factor of 5.1 (Fig. 1f). The IP:OP ratios of the sand, silt and clay size fraction amounted on average to 10.2, 2.9

5 and 2.0 (Fig. 1f), indicating that on average 73%, 58% and 50%, respectively, of the P was present in inorganic form. The TOP concentration was strongly related to the TOC concentration (R<sup>2</sup>=0.80, p<0.001), and the mean molar TOC:TOP ratio amounted to 250. Similarly, the OP concentration was significantly correlated to the OC concentration of the sand, silt and clay size fraction (R<sup>2</sup>=0.49, 0.70, 0.61, respectively, all p<0.001).</p>

The NMR data showed that the PME:PDE ratio in the topsoils ranged on average between 3.0 and 4.8 across all particle size fractions, indicating a general dominance of PME over PDE in all particle size fractions (Fig. 2). The PME:PDE ratio of the clay size fraction was significantly (p<0.01) lower than the PME:PDE ratio of the silt size fraction, while it was only marginally significantly (p<0.08) lower than the PME:PDE ratio of the sand size fraction (Fig. 2). However, if only the data of the temperate zone were considered, the PME:PDE ratio of the clay size fraction was significantly (p<0.001) smaller than the one of both the silt and the sand size fraction.</p>

## 3.2 Global distribution of C and P in particle size fractions

TOP was mainly correlated with MAT ( $R^2=0.47$ , p<0.001), whereas TOC was mainly correlated with MAP ( $R^2=0.42$ , p<0.001; Table 2, Supplement 3). Both TOP and TOC were correlated with the sum of latitude, MAT and MAP ( $R^2=0.69$  and 0.71, respectively, both p<0.001). TIP was also correlated with MAP ( $R^2=0.25$ , p<0.001) but not with latitude and MAT, while TP was not significantly (p>0.05) correlated with latitude and MAT, either,

and only weakly with MAP (Table 2).

The OP concentrations of the silt size and clay size fraction were both most strongly correlated with MAT ( $R^2$ =0.30 and 0.31, respectively, both p<0.001; Fig. 3a and b), similar to the TOP concentrations. The OC concentrations of the clay size fraction were also most strongly correlated with MAT ( $R^2$ =0.48 p<0.001; Fig. 3c), while the OC

- concentrations of the sand and the silt size fraction were more strongly correlated with MAP (R<sup>2</sup>=0.45 and 0.31, respectively, both p<0.001), similar to TOC. Much of the variability of the OC concentration of the clay size fraction was explained by the combination of latitude, MAT and MAP (R<sup>2</sup>=0.73, p<0.001; Table 2). In contrast to OP, the IP concentrations of the clay size and silt size fraction were not significantly (p>0.05) correlated with the MAP, and significantly but less strongly than OP with MAT and latitude (Table 2). The P concentrations (sum of the Clay size and silt size fraction were not significantly (p>0.05) correlated with the MAP, and significantly but less strongly than OP with MAT and latitude (Table 2). The P concentrations (sum of the clay size and silt size fraction were not significantly but less strongly than OP with MAT and latitude (Table 2). The P concentrations (sum of the clay size and silt size fraction were not significantly but less strongly than OP with MAT and latitude (Table 2). The P concentrations (sum of the clay size and silt size fraction were not significantly but less strongly than OP with MAT and latitude (Table 2). The P concentrations (sum of the clay size and silt size fraction were not significantly fractions (sum of the clay size and silt size fractions (sum of the clay size and silt size fractions (sum of the clay size and silt size fractions (sum of the clay size and silt size fractions (sum of the clay size and silt size fractions (sum of the clay size and silt size fractions (sum of the clay size and silt size fractions (sum of the clay size and silt size fractions (sum of the clay size and silt size fractions (sum of the clay size and silt size fractions (sum of the clay size and silt size fractions (sum of the clay size and silt size fractions (sum of the clay size and silt size fractions (sum of the clay size and silt size fractions (sum of the clay size and silt size fractions (sum of the clay size and silt size fractions (sum of the clay size an
- OP and IP) of the particle size fractions showed no correlation with latitude, MAT or MAP, and only a weak correlation with a combination of the variables for the clay size fraction (Table 2).

The TOC:TOP ratio was most strongly correlated with latitude ( $R^2$ =0.20, p<0.001; Table 3). The OC:OP ratios of the sand, silt and clay size fraction were also mainly correlated with latitude ( $R^2$ =0.22, 0.49 and 0.34, respectively, all p<0.001; Fig. 3e and f, Table 3). The OC:OP ratio of the clay size fraction changed less strongly with latitude

than the OC:OP ratio of the silt and the sand size fraction, as indicated by the slopes of the linear regression models, which amounted to -40.8, -10.4, and -3.7, in the sand, silt and clay size fraction, respectively (Figure 3e and f, Supplement 3). The TIP:TOP ratio and the IP:OP ratio of the clay size fraction were both correlated with MAT (R<sup>2</sup>=0.32 and 0.34, respectively, both p<0.001; Table 3).</p>


#### 3.3 Effects of land use change on C and P

Due to conversion from (semi-)natural vegetation to cropland, only the TOC (p<0.05) but not the TOP, TIP and TP concentrations (all p>0.05) changed significantly in the topsoils (Fig. 4a). The change in TOC amounted to -58 (Fig. 4a). The changes in TOP TIP and TP concentrations amounted to -35%, +23% and -11%, but were not statistically significant (4a).

The OC concentrations of all three particle size fractions were significantly (p<0.05) lower in the croplands than in the soils under (semi-)natural vegetation (Fig. 4b). The OC concentration decreased significantly (p<0.05) in the sand, silt and clay size fraction by 71, 47 and 35%, respectively (Fig. 4b). In contrast to OC, the OP concentration was only significantly (p<0.05) decreased in the sand, but not in the silt or in the clay size fraction

due to land use conversion from native to arable land and the change in the sand size fraction amounted to -70% (Fig. 4c).

In contrast to OC and OP, the concentrations of IP in the particle size fractions rather increased due to land use change (although not significantly). No significant (p>0.05) difference between the particle size fractions was found for IP (Fig. 4d). Similarly, the P concentrations (sum of organic and inorganic P) of the particle size fractions

did not change significantly (p>0.05) with land-use change, and the relative changes in the P concentrations of the particle size fractions did not differ significantly (p>0.05) from each other, either (data not shown).

#### 3.4 Vertical distribution of C and P

In the soil profiles, the OP concentration of the sand size fraction was very small throughout all soil depths, while
the OP concentrations of the silt and especially of the clay size fraction were much larger in the upper 35 cm (Fig. 5a) similar to the topsoils (Fig. 1). Below 35 cm, the OP concentrations of the silt and the clay size fraction decreased strongly (Fig. 5a). The ratio of IP:OP was much higher in the sand than in the silt and in the clay size fraction in the upper 50 cm (Fig. 5b). However, with increasing depth, the IP:OP ratio increased in the clay and especially in the silt size fraction as indicated by the slopes of the liner models that amounted to 0.56 and 0.19, respectively (Fig. 5b).

## 4. Discussion

#### 4.1 Distribution of OP and OC among particle size fractions

The OC:OP ratios of the sand, silt and clay size fraction equaled 771, 424, and 204 (Fig. 1e), and were much lower 30 than the C:P ratios of senesced plant leafs and leaf litter, which on a global average amount to 1183 and 1334 (McGroddy et al., 2004; Yuan and Chen, 2009). Thus, OP is strongly enriched in soil compared to plant detritus, especially in the clay size fraction. The reason for this is likely that OP sorbs more strongly to the soil mineral phase than non-phosphorylated organic compounds, which leads to physical protection of OP against microbial decomposition, and thus to an enrichment of OP in soils. Hence, organic phosphorylated compounds seem to be

much more persistent in soil than non-phosphorylated organic compounds.

The reason for why the OP concentration was significantly higher in the clay size fraction than in the silt size fraction, and higher in the silt size fraction than in the sand size fraction in the topsoils (Fig. 1a) and in the subsoils (Fig. 5a) is likely the different mineralogy of the fractions. The clay size fraction is rich in Fe and Al oxides and hydroxides, and the silt fraction also contains a higher proportion of Fe and Al oxides and hydroxides than the

5 sand size fraction, which is dominated by quartz (Sposito et al., 1999; Christensen et al., 2001; von Lützow et al., 2006). Thus, the silt and especially the clay size fraction strongly sorb OP due to the positive charge of the Fe and Al oxides and hydroxides.

The finding that OP was much stronger enriched in the clay size fraction (compared to the sand size fraction) than OC and also stronger than IP (Fig. 1) indicates that OP has a higher capacity to compete for binding sites in the

- clay size fraction than OC and IP. The most likely explanation for this is that OP is dominated by inositol phosphates which has multiple phosphate groups, and thus a higher capacity to compete for binding sites than IP (McKercher and Anderson, 1989; Martin et al., 2004; Celi and Barberis, 2005; Ruttenberg and Sulak, 2011). IP was less strongly enriched in the clay size fraction (compared to the sand size fraction) than OP, and more evenly distributed among all three particle size fractions, leading to significantly increased IP:OP ratios in the sand size
- fraction (Fig. 1f), which underlines that a large proportion of the OP had a higher capacity than IP to compete for binding sites in the two smaller particle size fractions. However; in the subsoil, the IP:OP ratio increased with soil depth (Fig. 5b) as OP decreased (Fig. 5a). Fine size particles are more prone to get eroded than larger particles. Thus, the finding that OP and IP are enriched in the clay size particle suggests that losses of fine textured material likely lead to comparatively high P losses.
- The low OC:OP ratio of the clay size fraction shows that the OM in this fraction has been strongly decomposed since the C:P ratio of OM decreases during decomposition (Blair, 1986; Berg and McClaugherty, 1989; Manzoni et al., 2010; Spohn and Chodak, 2015). Since decomposition of OM requires time this might indicate that the OM in the clay size fraction is on average older than the OM in the two other particle size fraction (Spohn and Sierra, 2018) as indicated by previous studies (Anderson & Paul, 1984; Scharpenseel & Becker-Heidmann, 1989; Quideau
- et al., 2001; Eusterhues et al., 2003). On the other hand, the high P content of the OM in the clay size fraction might stabilize OM due to sorption of the phosphate groups to mineral surfaces, leading to further persistence of the OM in this particle size fraction.

The larger proportion of PDE in the clays size fraction compared to the silt size fraction (Fig. 2) can be explained by the fact that Al and Fe oxides, which largely form the mineral phase of the clay size fraction, strongly sorb OP and also stabilize the more labile PDE (Tate, 1984; Steward and Tiessen, 1987; Darch et al., 2014). In other words, despite their relatively low capacity to compete for sorption sites, PDEs persist longer in the clay size fraction than in the other two particle size fractions due to the high concentration of Al and Fe oxides in the clay fraction that strongly sorb OP. Yet, PME were dominating in all particle size fractions, which is in accordance with a general dominance of PME reported in a meta-analysis on OP in fertilizers, soils and waters (Darch et al., 2014) and in a

35 review on OP in tropical soils (Nziguheba and Bünemann, 2005). Furthermore, the result is in accordance with von Sperber et al. (2017), indicating that the PME:PDE ratio does not change during to cultivation.

#### 4.2 Global distribution of OC and OP in particle size fractions

This is the first study, to my knowledge, to show that the OP concentration of particle size fractions decreases with increasing MAT and that the OC:OP ratio of the particle size fractions increase with latitude (Fig. 3, Table 2 and 3). The reason for the decrease in OP with increasing MAT is likely that the P concentration of plant detritus decreases with increasing MAT (McGroddy et al., 2004; Yuan and Chen, 2009). The reason for the decrease in

- decreases with increasing MAT (McGroddy et al., 2004; Yuan and Chen, 2009). The reason for the decrease in the P concentration of plant leaf detritus, in turn, is a decrease in the P concentration of plant leafs in combination with an increase in plant P resorption from senescent leafs (before leaf abscission) with increasing MAT (McGroddy et al., 2004; Yuan and Chen, 2009). The former ultimately results from an elevation in plant productivity with increasing MAT in combination with P limitation caused by the high weathering rates under
- warm climate (McGroddy et al., 2004; Reich and Oleksyn, 2004; Vitousek and Sanford, 1986). In contrast to TOP, the TOC concentration of the bulk soil and the OC concentrations of the sand and silt size fraction were most strongly correlated with MAP (Table 2). This is in accordance with a meta-analysis on soil TOC contents and can be attributed to the fact that vegetation type and plant productivity are strongly related to precipitation (Jobbágy and Jackson, 2000). As a result of the decrease in OP with decreasing MAT and the increase in OC with increasing

MAP, the OC:OP ratios of the particle size fractions increased with latitude (Fig. 3, Table 3).

The finding that the OC:OP ratio changed less with latitude in the clay than in the silt and sand size fraction (Figure 3e and f, Table 3) can be attributed to the high sorption capacity of the clay size fraction that seems to partially compensate for low OP inputs at low latitudes. The correlations found for the OC:OP ratios of the particle size fractions, especially of the clay and the silt size fraction, are stronger than the correlations found for the TOC:TOP

- ratio of the bulk soil (Table 3). This indicates that the relationships between latitude and soil stoichiometry are masked in the bulk soil due to the variability of the masses of the particle size fractions. Thus, the results suggest that particle size fractionation has a large potential to render relationships between climate and soil stoichiometry visible that are obscured in bulk soil by differences in soil texture. While the relationship between latitude and OC:OP ratios of the particle size fractions has not yet been revealed, to my knowledge, the global molar TOC:TOP
- ratio of the bulk soil found here, which amounted to 250, is in agreement with data presented in Kirkby et al. (2011).

In contrast to OP and OC, the TIP of the bulk soil and the IP concentrations of the particle size fractions were less strongly correlated with latitude, MAT and MAP (Table 2). The reason for the stronger dependence of OC and OP on latitude and climate is likely that biomass production clearly depends on climate, and thus on latitude, whereas

IP and total P concentrations depend more firmly on the P content of the bedrock (Porder and Ramachandran, 2013). In contrast to the findings presented here, Siebers et al. (2017) found a strong correlation between TP and MAT in soils in the USA. The reason for this apparent contradiction is likely that the parent materials of the soils in Siebers et al. (2017) were more similar among each other and did not cover such a wide spectrum as in the present global meta-analysis. Hence, at smaller spatial scales, and in areas with similar bedrock, TP might also be affected by MAT.

#### 4.3 Effect of land-use change on C and P

The finding that the TOC but not the TOP concentration changed significantly due to land use conversion from (semi-)native vegetation to cropland in the topsoils (Fig. 4) indicates that TOP is more persistent in soil than TOC. This is likely due to sorptive stabilization of organic phosphorylated compounds (see above). Most land-use

- changes considered here were changes from grassland to cropland (Table 1), and the TOC concentration in the topsoils decreased on average by 58% (Figure 4a). Thus, the data presented here are in accordance with a global meta-analysis on land-use change, reporting that the conversion of grassland to cropland leads to a decrease by 59% and the conversion of forest to cropland leads to a loss of 42% of the initial TOC in the topsoils (Guo and Gifford, 2002). This agreement indicates that the changes in TOC found here are very close to the global mean
- calculated based on a much larger number of observations, despite the fact that only ten land use comparisons were considered here. The differences in OC concentrations between the cropped soils and the native soils were significant for all three particle size fractions, and even in the clay size fraction, a significant share (35%) of the OC was lost due to land-use conversion (Fig. 4b). In contrast, the OP concentration changed only significantly in the sand size fraction due to land use change (Fig. 4c), suggesting that OP in the silt and in the clay size fraction
- is more strongly protected against microbial decomposition than OC. This provides strong support for the idea that many OP compounds compete more successfully for sorption sites in these two particle size fractions than nonphosphorylated organic compounds.

The result that the TIP concentration (Fig. 4a) as well as the IP concentrations of the particle size fractions (Fig. 4d) rather tended to increase (although not significantly) due to land use conversion suggests that the majority of

20 OP compounds were not leached out of the soil, but were mineralized. Leaching of OP might also have been compensated by inputs of IP fertilizer at some of the sites. However, all in all, the small (and not significant) decrease in the TP (Fig. 4a) indicates that the conversion of sites with native vegetation to croplands in general rather evokes a decrease in the TP contents, which can likely be attributed to an increased plant P uptake at the cropped sites in combination with biomass removal (MacDonald et al., 2011).

#### 4.4 Conclusions and implications

By studying (i) the distribution of OP and OC among particle size fractions in topsoils and subsoils, (ii) the global distribution of OC and OP in particle size fractions, and (iii) the effect of land-use change on OC and OP in particle size fractions, it was found that OP is more persistent in soil than OC, which is likely due to the fact that OP competes more successfully for sorption sites in soil, and especially in the clay size fraction. Strong sorption makes soil OP less vulnerable to land-use change than OC. In the clay size fraction, the strong sorption seems to compensate for latitude-dependent differences in inputs of OP. The study indicates that OP might play an important role in the formation of organo-mineral complexes, which should be further investigated in the future. In addition, the meta-analysis showed that particle size fraction has a large potential to render relationships between

35 climate and soil stoichiometry visible that are masked by differences in soil texture, and thus are not apparent form the analysis of bulk soil. The results of this study have important implications as they suggest that soil OP is very strongly sorbed to soil minerals and is less susceptible to changes in land use and MAT than soil OC. This is positive in the sense that OP is not as vulnerable as OC with respect to global change. However, the results also

suggest that it will be very difficult to substitute expected future shortages of P fertilizer, even on the short run, by soil OP since a large share of OP is strongly protected against mineralization and plant uptake by sorption.

## Data availability

All data is available in the supplement.

#### Acknowledgments

MS thanks the authors of all studies that were considered in this meta-analysis for their work and the German Research Foundation for funding through the Emmy Noether-program (grant SP1389/6-1).

#### 10

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
