# Peer review of "Phosphorus and carbon in soil particle size fractions – A global synthesis"

_Biogeosciences, 2018_

## Referee Comment (RC1) · Anonymous Referee #1 · 2 Nov 2018

With an increasing number of studies on the fate and origin of soil P, a meta-analysis of published data is principally welcome. Yet, this paper hardly adds novel findings to what has been published, it ignores methodological differences among studies, and does not really go into depth with statistical analyses, which in this form are even wrong. I therefore have majors concerns regarding the publication of this paper in its present form.

The main criticism refers to:

Lacking novelty: There are numerous studies on the effect of land use on OP and IP in soil, and the conclusions drawn here do not add much to current process under-standing. Also, correlations to climatic elements have been reported earlier and are no achievement of the current analyses. Generally, the discussion sections fails to make

clear, which findings have been reported and discussed earlier in individual papers, and what is the additional merit when summarizing them.

Inadequate literature research and number of studies considered: This paper has 49 references, lacking quite a few publications on the related topic. The "meta-analysis" itself relies on 11 studies only (Table 1); even as I can understand that the authors wished to have all analyses within the same study, this is by far too small to discuss global variations in OP and IP distribution in relation to OM and other site indices. Searching for the combination of soil organic matter AND particle size fraction in Chemical Abstracts, for instance, already provides 842 hits. Further combining with phosphorus provides 84 hits, far more than considered here in this paper. One of the best reviews on organic matter in size separates was published by Christensen (1992; Adv. Soil Sci) –this review 26 years ago already had more references than this meta-analysis – which is not even cited in this study!

Statistical analyses inadequate: The author uses ANOVA to compare different sampling sets and transforms data for normality. Yet, i) ANOVA relies on independent data while the size fractions considered here are dependent data. If independent data sets would remain, ii) the main problem in ANOVA is inhomogeneity of variances, which has to be tested, but no info on this are included in the manuscript. Third, all conclusions can be made for transformed data only but not for original ones. Fourth, ratios are per-se not normally distributed, i.e., comparisons have to consider geometric means rather than arithmetic ones etc. Fifth, despite the author stating that some data had to be transformed to fit to normality, they perform linear regression analyses only. Sixth, in none of the statistical treatments they account for covariate interactions. And seventh, no tests were performed on the stability of statistical analyses. Overall, this is not sufficient for a high-level journal such as Biogeosciences.

Statistical parameter selection questionable: the author states to find novel insights by correlating P data to latitude. There are several issues here. Correlations and regressions that improve by additionally including latitude might be spurious, as latitude

should be highly correlated with MAT at sea level. On the other hand, looking at the study by Makarov et al. 2004, all sites have the same latitude, but across altitudes there is considerable variation in MAT and vegetation. In the discussion, latitude is at times used interchangeably with climate, which at least for the study by Makarov is not true. For the same reason, it is not clear which sites you included in which climate zone (e.g., p.6 l.13/14). Further, many other site properties change with latitude, such as geology, interactions with vegetation type, land-use system etc., all of these have not been considered in statistical analyses.

Ignorance of classification systems: The author just compares the concentrations of OP and IP in clay, silt and sand, apparently ignoring that different countries use different size thresholds for silt and sand, e.g., $20\mu$m according to ISSS and Australia, 50 or $53\mu$m in France or US, 63 $\mu$m in Germany.

Differences in methodology not considered: The NMR extraction methods changed in the last 20 years. How was this considered here? Also, extractability of P possibly affects NMR results, which was fully ignored in data evaluation here. Similar problems hold true for OP data. These were also obtained from pooling results of Hedley fractions. Which fractions were pooled and were they the same for all studies? Which were the criteria for OC data to be included in the study (methods etc.)? Besides, also the methodologies used for aggregate dispersion prior to particle-size fractionation likely varied between studies, which was also not discussed by the author.

P stocks not analyzed: Land-use effects on soil P should not be considered on a soil concentration but on a total P stock basis, i.e., by multiplying OP and IP concentrations with bulk density and mass of the fractions. This, however, has not been done. Yet, it might be important, as already Christensen et al. (1992; Adv. Soil Sci) stated that element enrichment within a size fraction is non-linearly, inversely proportional to clay content.

Simplified discussion: Several statements in the introduction and discussion section

are very simplified or mere speculation, not justified by research. For instance:

- I doubt that information on OC in soils is limited (p.2, l. 21),

- the author calculates molar ratios ( (p. 4, l. 31) without knowing molecular weights,

- IP losses during size fractionation or incomplete extractability for NMR was not considered in the results section;

- both introduction and discussion are too strongly focused on the sorptive strength of the phosphate group as the only parameter explaining changes in OC:OP ratios and OP distribution into size fractions;

- lower concentration changes in soil P than of soil C may have nothing to do with persistence, but merely indicate that there is no significant gaseous P loss pathway as opposed to soil C, and that farmers fertilize P to compensate for these losses;

- the observed shift in IP:OP ratio with depth is a result of decreasing OM contents with depth and simultaneously increasing proportions of P containing minerals which are only partially weathered or not at all. This is, however, completely unrelated to the higher sorption strength of OP compared to IP or the fact that fine particles are eroded more easily as it is currently framed in the discussion (p.8, l. 15-20);

- If there was such a large range in particle-size distributions and other parameters for 11 studies only (p. 9, l. 21 and 33-34), how does this affect the results? Merely stating that this likely blurred the findings (p. 9, l. 21) is unsatisfying for a meta-analysis;

- I doubt that there was significant OP leaching – the authors should estimate cumulative leaching rates (not done) and related them to P stock changes (not calculated) before speculating that this process was relevant (without even citing related studies);

- there are sections in the introduction and discussion linking P in soils to P in vegetation and plant litter, but this is barely considered in the analysis.

Redundancy of several parts: e.g. p. 1, l. 1-27: hardly introduce into the objectives;

p.2, l. 12-22: no connection to former paragraph; p. 6, l. 22-32: what is novel here compared to correlations shown before; p. 7, l. 18-26: well known, p. 8, l. 1-18: well known; p. 9: this is not the first study relating P to climate; p. 10, l. 26-37: Conclusions: too simplified, partly well known; p. 11, l. 1-2: mere speculation, no data have been provided on sorptive stabilization of SOP against degradation.

Poor technical presentation of the manuscript: This manuscript should have been checked by an independent corrector for punctuation, spelling and style as there are a number of mistakes, ranging from mere "typos" to completely wrong (e.g. abbreviations in table headings).

---

## Referee Comment (RC2) · Anonymous Referee #2 · 12 Nov 2018

General comments

The study encompasses a relevant and interesting topic and a review on this topic would be a good addition to the existing literature. However, the number of studies involved is relatively small to draw conclusions on a global scale, especially because some of the dominating drivers of the differences among the included studies are not (or cannot be) taken into account or addressed/discussed:

- Differences in horizon and sampling depth are not taken into account or addressed for the global meta-analysis. This can have large consequences for the total and relative amount adsorbed to different soil fractions, as both carbon and P tend to accumulate in the top few cm of the soil, as shown in figure 5a.

- Differences in land uses are only addressed in the section on land use change, but are very likely to be a main driver of the differences in the entire dataset. Phosphorus (and carbon) inputs widely differ among agricultural and natural systems, and this should be addressed (more prominently) in the current study.

- The effect of soil pH is not addressed. The author mentions adsorption of (organic) P to metal (hydr)oxides as one of the main drivers. The affinity of different P compounds to adsorb to reactive surfaces strongly depends on pH, as the author mentions in the introduction. This point does not receive any further attention in the rest of the paper.

Considering that these factors remain largely unaddressed and taking into account the limited size of the analysis (only 11 studies were incorporated), I think the manuscript is unfit for publication in its current form.

Additionally, the manuscript is not written very elegantly. There are many repetitions, errors, and grammatically incorrect sentences throughout the manuscript that give it an unfinished look.

Specific comments

Introduction:

- The introduction can be written shorter and more concisely towards the goal of the study. Some examples:

- The first two paragraphs (p. 1, l. 5) can be summarized in two or three short sentences that indicate P is important for organisms and distinguishing between IP and OP in soil.

- The information on different inositol phosphates (p. 1, l. 18) is not relevant here.

- The entire paragraph on sorption of P monoesters and diesters might be replaced with one or two sentences on the difference in sorption affinity between higher-order inositol phosphates, P diesters and inorganic P (orthophosphate). The last remark on competition between inorganic P and organic carbon comes out of the blue and does

not serve any purpose here.

- The section on the C:P ratio of the plant input (p. 2, l. 11) can be shortened. There is no need to discuss the results of these meta-analyses in detail. Additionally, in l. 14 the author suggests a decreasing order, but the signs suggest an increasing order.

- On p. 2, l. 22 the author describes the objectives of the study, but the effect of soil depth and land use are not mentioned anywhere in the introduction.

- When the effect of P inputs on OC/OP ratio is discussed (p. 3, l. 11), the author mentions plant material, but not other inputs like (organic or mineral) P fertilizer or animal waste/manure. This would be acceptable if the meta-analysis was restricted to forest ecosystems, but that is not the case.

Materials and Methods:

- The search method is not reproducible. I would expect a search query.

- It is not clear to me why studies that report only two pools of organic P in the Hedley fractionation are included?

- Saunders and Williams describe several methods in their paper. I assume the author is referring to the ignition method.

- In the Saunders and Williams ignition method, it is organic P that is determined as the difference between total and inorganic P, not the other way around as the author states on page 4.

- Both methods to determine the soil P fractions have their pros and cons, yet the influence of the procedure or the possible implications of including different methods are not discussed. I would expect a section on this topic or an analysis to the effect of the method used in the studies, as it can significantly affect the results and interpretation.

- Not enough information is provided on the NMR analysis. NMR analysis results may depend on many variables, including the type of NMR (solid vs. liquid state), the extraction procedure used, delay and acquisition times, etc.

Results and Discussion:

- In general, there is not enough comparison to the existing literature on the implications of the findings.

- The soil clay fraction usually contains more Fe and Al oxides as mentioned in the discussion. It is a shame there is no information available on the Fe and Al content of the soils in the studies, as this could have helped strengthen this point. Adsorption to clay should not be overlooked here either, as clay minerals can substantially contribute to the reactive surface area of a soil (see Gerard et al. 2016 Geoderma 262:213-226)

- The conclusion about increased P losses from erosion (p. 8, l. 17) looks a little odd, without any context. The fact that the soil clay fraction is richer in organic material and nutrients is well known, so this conclusion is of limited novelty.

- I am not convinced that the lower OC:OP ratio in the clay size points towards a further degraded OM source (P. 8, l. 20). It might indicate that it contains OM of a different origin, or a preferential adsorption of organic compounds with a relatively high P content.

- The ratio between P monoesters and diesters seems very low (relatively high abundance of diesters) compared to other previous analyses (e.g. Menezes-Blackburn et al. 2018 Plant & Soil 427:5-16). I am wondering why this is the case.

- It seems to me that latitude and MAT should be strongly correlated. Yet their relation to the measured P pools is dissimilar. This topic should be disentangled and explained further.

- Land-use change had no effect on P stocks in this study. The data do thus not support the suggestion made in the last sentences on this section (p. 10, l. 21). As data on additional inputs (fertilizer, manure) and outputs (crop yield) of C and P are not provided, any conclusions on mineralization or leaching of organic P are purely

speculative and should be avoided.

- In the conclusion, the author makes a remark about the potential of soil OP to replace P fertilizer inputs. This comes out of the blue and should be discussed in a discussion section first.

---

## Author Comment (AC2) · 23 Nov 2018

The response has been uplaoded as a supplement.

Please also note the supplement to this comment:
https://www.biogeosciences-discuss.net/bg-2018-404/bg-2018-404-AC2-supplement.pdf

---

## Author Comment (AC3) · 23 Nov 2018

**Response by Marie Spohn**

I thank the two referees for reviewing the manuscript. Some of the comments helped me to improve the manuscript. However, other comments are not justified.

A main concern of both referees is the number of observations on which this meta-analysis is based. I found 118 observations (data on 118 soil horizons) that match the criteria of my literature search. Thus, the size the meta-analysis is comparable to other meta-analyses on phosphorus (P) in soils. The meta-analysis by Cross and Schlesinger (1995) on soil Hedley P fractions, for example, is based on 88 soil horizons, and the meta-analysis by Yang and Post (2011) added 90 observations to the data set. More recently, a meta-analysis by Darch et al. (2014) analyzed NMR data of 18 soil horizons. Thus, I do not think that the concern that the study is based on too few observations is justified.

The referees are also concerned about differences in the methods used in the studies on which the meta-analysis is based. Anonymous referee#1 points out that extraction techniques for NMR analyses might differ among studies, that the definition of the sizes of the particle size fractions might differ and that the study sites differ in altitude. Anonymous referee#2's main concern is that the soil sampling depths differ among studies.

Most of the concerns are not justified. The NMR data have all been obtained by the same method (see below), only two different definitions of the threshold between the silt and the sand size fraction are used in the studies (see below). Further, the analysis of the vertical distribution of organic phosphorus (OP) shows that the concentrations differ not significantly in the upper 20 cm of the soils, thus differences between the sampling depth of the topsoils should not have a significant effect (see below).

Moreover, it is in the nature of a meta-analysis that it is based on studies that use slightly different methods. If in meta-analyses, we would only consider studies that use exactly the same methods, not a single synthesis paper could have been written in soil science and biogeochemistry. The criteria of my literature search ensure that the methods used in the single case studies are sufficiently similar to allow comparison of the results.

In the following, I will respond in detail to the comments.

**Anonymous Referee #1**

With an increasing number of studies on the fate and origin of soil P, a meta-analysis of published data is principally welcome. Yet, this paper hardly adds novel findings to what has been published, it ignores methodological differences among studies, and does not really go into depth with statistical analyses, which in this form are even wrong. I therefore have majors concerns regarding the publication of this paper in its present form.

The main criticism refers to:

Lacking novelty: There are numerous studies on the effect of land use on OP and IP in soil, and the conclusions drawn here do not add much to current process under-standing. Also, correlations to climatic elements have been reported earlier and are no achievement of the current analyses. Generally, the discussion sections fails to make clear, which findings have been reported and discussed earlier in individual papers, and what is the additional merit when summarizing them.

The reviewer claims that my study lacks novelty and that the "correlations to climatic elements have been reported earlier". I am not aware of any study that conducted a similar meta-analysis on OP and OC in soil particle fractions. The reviewer should name these studies. Otherwise, this remark seems to have no fundament at all.

Inadequate literature research and number of studies considered: This paper has 49 references, lacking quite a few publications on the related topic. The "meta-analysis" itself relies on 11 studies only (Table 1); even as I can understand that the authors wished to have all analyses within the same study, this is by far too small to discuss global variations in OP and IP distribution in relation to OM and other site indices. Searching for the combination of soil organic matter AND particle size fraction in Chemical Abstracts, for instance, already provides 842 hits. Further combining with phosphorus provides 84 hits, far more than considered here in this paper. One of the best reviews on organic matter in size separates was published by Christensen (1992; Adv. Soil Sci) –this review 26 years ago already had more references than this meta-analysis – which is not even cited in this study!

Based on the criteria of the literature search, data on OP in particle size fractions of 118 soil horizons were found. Thus, the size the meta-analysis is comparable to other meta-analyses on P in soils. The meta-analysis by Cross and Schlesinger (1995) on soil Hedley P fractions, for example, is based on 88 soil horizons, and the meta-analysis by Yang and Post (2011) added 90 observations to the data set. More recently, a meta-analysis by Darch et al. (2014) analyzed NMR data of 18 soil horizons.

I am confident that I included all peer-reviewed studies that match the criteria of the literature search. If the reviewer can name a single study that I ignored albeit the fact that it matches the criteria of the literature search, I am happy to include it in the study. The criteria of my literature search ensure that the methods used in the single case studies are sufficiently similar to allow comparison of the results. The review by Christensen (1992) is on OC in particle size fractions not on OP. Thus, many of the references therein do not match the criteria of the literature search. Still, I added the study as a reference in the Introduction.

Statistical analyses inadequate: The author uses ANOVA to compare different sampling sets and transforms data for normality. Yet, i) ANOVA relies on independent data while the size fractions considered here are dependent data. If independent data sets would remain, ii) the main problem in ANOVA is inhomogeneity of variances, which has to be tested, but no info on this are included

in the manuscript. Third, all conclusions can be made for transformed data only but not for original ones. Fourth, ratios are per-se not normally distributed, i.e., comparisons have to consider geometric means rather than arithmetic ones etc. Fifth, despite the author stating that some data had to be transformed to fit to normality, they perform only. Sixth, in none of the statistical treatments they account for covariate interactions. And seventh, no tests were performed on the stability of statistical analyses. Overall, this is not sufficient for a high-level journal such as Biogeosciences.

The reviewer is right in saying that the different particle size fractions of one sample are not normally distributed. Therefore, I removed the ANOVA results from Figures 1 and 2. Moreover, I also added the geometric mean of the ratios in Fig. 1e and f.

Statistical parameter selection questionable: the author states to find novel insights by correlating P data to latitude. There are several issues here. Correlations and regressions that improve by additionally including latitude might be spurious, as latitude should be highly correlated with MAT at sea level. On the other hand, looking at the study by Makarov et al. 2004, all sites have the same latitude, but across altitudes there is considerable variation in MAT and vegetation. In the discussion, latitude is at times used interchangeably with climate, which at least for the study by Makarov is not true. For the same reason, it is not clear which sites you included in which climate zone (e.g., p.6 I.13/14). Further, many other site properties change with latitude, such as geology, interactions with vegetation type, land-use system etc., all of these have not been considered in statistical analyses.

It is in the nature of a meta-analysis that it is based on data of soils from different altitudes. The correlations of OP and OC with mean annual temperature (MAT) and mean annual temperature (MAT) are discussed separately at length in section 4.2. The effect of land-use is discussed in a whole section (section 4.3), and an analysis on the effect of land-use conversion on OP and IP in particle size fractions has been conducted (Figure 4). The effects of vegetation and geology are discussed at the beginning of section 4.1. Therefore, I think that these remarks by the reviewer have no fundament.

I removed the results of the multiple regressions that contained both latitude and MAT or latitude and MAP as independent variables.

Ignorance of classification systems: The author just compares the concentrations of OP and IP in clay, silt and sand, apparently ignoring that different countries use different size thresholds for silt and sand, e.g., 20\_m according to ISSS and Australia, 50 or 53\_m in France or US, 63 \_m in Germany.

**All studies here used either 20 or 50 $\mu$ m as a threshold for the silt and the sand size particle fraction. I added a table to the supplement that shows the definition of the size of each particle size fraction in each study.**

Differences in methodology not considered: The NMR extraction methods changed in the last 20 years. How was this considered here? Also, extractability of P possibly affects NMR results, which was fully ignored in data evaluation here. Similar problems hold true for OP data. These were also obtained from pooling results of Hedley fractions.

All NMR data were obtained using the same method. Soils were extracted in diluted NaOH solution, the extracts were centrifuged and dialyzed, and finally measured on a NMR spectrometer (Bruker Instruments). All NMR analyses were conducted around the year 2000 in the same lab in

**Germany and all papers providing NMR data were published within only five years. Thus, I think that the concern raised here by the reviewer has no fundament.**

Which fractions were pooled and were they the same for all studies? Which were the criteria for OC data to be included in the study (methods etc.)? Besides, also the methodologies used for aggregate dispersion prior to particle-size fractionation likely varied between studies, which was also not discussed by the author.

The answer to the first question is given in the method section. There was no criterion for OC in the literature search. The studies were selected based on the criteria for P because much less studies report data on P than on C in particle size fractions. All studies on which the meta-analysis is based that report OC data measured it using a CN analyzer.

P stocks not analyzed: Land-use effects on soil P should not be considered on a soil concentration but on a total P stock basis, i.e., by multiplying OP and IP concentrations with bulk density and mass of the fractions. This, however, has not been done. Yet, it might be important, as already Christensen et al. (1992; Adv. Soil Sci) stated that element enrichment within a size fraction is non-linearly, inversely proportional to clay content.

I agree with the reviewer that it would be good to also analyze the stocks of the bulk soil, and I added a panel showing the change in TOC, TOP, TIP, and TP stocks due to land-use conversion to Figure 4.

**Figure X** Changes in stocks of TOC, TOP, TIP and TP in the bulk soil due to land use change from (semi-)native vegetation to cropland. Numbers above each boxplot depict the mean  $\pm$  the standard deviation. The mean is also indicated by a red square. Different capital letters indicate significant differences between changes in TOC, TOP and TIP in the bulk soil.

The idea behind integrating data on land-use change from (semi-)natural vegetation to cropland is to test how this disturbance affects the distribution of OP, OC and IP among particle size fractions with the aim to learn about the sequestration of OP, OC and IP in the three different particle size fractions. The effect of land use-change on OP, OC and IP in the three particle size fractions would be masked by differences in texture among the different soils if the concentrations were multiplied with the size of the fractions.

Simplified discussion: Several statements in the introduction and discussion section are very simplified or mere speculation, not justified by research. For instance:

- I doubt that information on OC in soils is limited (p.2, l. 21).

Apparently the reviewer got something wrong here. The sentence on page 2, line 21 says. "In contrast, PDEs have a lower charge density than orthophosphate and PME and their phosphate groups are considerably shielded from ionic interactions." The word limited only appears in this review but not in the manuscript.

- the author calculates molar ratios ( (p. 4, l. 31) without knowing molecular weights

I do, of course, know the molecular weight of C and P, and thus, I can calculate molar ratios of C and P.

- IP losses during size fractionation or incomplete extractability for NMR was not considered in the results section;

All NMR data were obtained using the same method around the year 2000 in the same lab in Germany. Thus, differences in the data caused by differences in the methods should be very small. An analysis of the loss of IP in the preparation of the samples for the NMR analyses is beyond the scope of this study which focuses on global patterns of OP and OC in particle size fractions (the NMR data is only a small part of this study).

- both introduction and discussion are too strongly focused on the sorptive strength of the phosphate group as the only parameter explaining changes in OC:OP ratios and OP distribution into size fractions;

I agree and I added that OP might also be more persistent than OC in soil due to the recalcitrance of some phosphorylated organic compounds.

- lower concentration changes in soil P than of soil C may have nothing to do with persistence, but merely indicate that there is no significant gaseous P loss pathway as opposed to soil C, and that farmers fertilize P to compensate for these losses;

**Lower concentration changes in soil OP (not total P!) than of OC might indeed indicate that OP is more persistent than OC in soil**

- the observed shift in IP:OP ratio with depth is a result of decreasing OM contents with depth and simultaneously increasing proportions of P containing minerals which are only partially weathered or not at all. This is, however, completely unrelated to the higher sorption strength of OP compared to IP or the fact that fine particles are eroded more easily as it is currently framed in the discussion (p.8, I. 15-20);

I agree with the reviewer, and the manuscript does as well since the section the reviewer refers to starts as follows. "in the subsoil, the IP:OP ratio increased with soil depth (Fig. 5b) as OP decreased (Fig. 5a)."

- If there was such a large range in particle-size distributions and other parameters for 11 studies only (p. 9, I. 21 and 33-34), how does this affect the results? Merely stating that this likely blurred the findings (p. 9, I. 21) is unsatisfying for a meta-analysis;

Here, the reviewer takes sentences out of their context. The whole paragraph says "The correlations found for the OC:OP ratios of the particle size fractions, especially of the clay and the silt size fraction, are stronger than the correlations found for the TOC:TOP ratio of the bulk soil (Table 3). This indicates that the relationships between latitude and soil stoichiometry are masked in the bulk soil due to the variability of the masses of the particle size fractions." (p. 9, I. 18-21).

- I doubt that there was significant OP leaching – the authors should estimate cumulative leaching rates (not done) and related them to P stock changes (not calculated)before speculating that this process was relevant (without even citing related studies);

I removed this sentence from the revised version of the manuscript.

- there are sections in the introduction and discussion linking P in soils to P in vegetation and plant litter, but this is barely considered in the analysis.

**This statement has no fundament. The relationship between C:P ratios in leaf litter and OC:OP ratios in soil is discussed in section 4.2.**

Redundancy of several parts: e.g. p. 1, l. 1-27: hardly introduce into the objectives;p.2, l. 12-22: no connection to former paragraph; p. 6, l. 22-32: what is novel here compared to correlations shown before; p. 7, l. 18-26: well known, p. 8, l. 1-18: well known; p. 9: this is not the first study relating P to climate; p. 10, l. 26-37: Conclusions: too simplified, partly well known; p. 11, l. 1-2: mere speculation, no data have been provided on sorptive stabilization of SOP against degradation.

I checked for redundancies in the discussion, and removed some sentences.

Poor technical presentation of the manuscript: This manuscript should have been checked by an independent corrector for punctuation, spelling and style as there are a number of mistakes, ranging from mere "typos" to completely wrong (e.g. abbreviations in table headings).

I did not find the "completely wrong abbreviations", but I will ask a native English speaker to look for them.

**Anonymous Referee #2**

**General comments**

The study encompasses a relevant and interesting topic and a review on this topic would be a good addition to the existing literature. However, the number of studies involved is relatively small to draw conclusions on a global scale, especially because some of the dominating drivers of the differences among the included studies are not (or cannot be) taken into account or addressed/discussed:

- Differences in horizon and sampling depth are not taken into account or addressed for the global meta-analysis. This can have large consequences for the total and relative amount adsorbed to different soil fractions, as both carbon and P tend to accumulate in the top few cm of the soil, as shown in figure 5a.

I added a table to the supplement in which the sampling depth of the topsoil is given for each study. The weighted mean depth of all topsoil horizons amounted to 15.0 cm. Furthermore, I added the following lines to the Results:

"The OP concentration in the depth segment 0-10 cm amounted on average to 110, 474, and 1030 mg kg-1 in the sand, silt and clay size fraction, respectively. The OP concentrations in the depth segment 10-20 cm were very similar and amounted to 66, 454, and 1041 mg kg-1 in the sand, silt and clay size fraction, respectively."

In addition, I added a section entitled "Methodological Considerations" at the beginning of the Discussion, which contains the following lines.

"As in any meta-analysis in soils science, the soil sampling depth differs between primary studies. This study analyzed the vertical distribution of OP in soils (Fig. 5), and found that the concentration of OP in the particle size fraction in the top 10 cm did hardly differ from the depth segment ranging from 0.1 to 0.2 m. Thus, the error derived from the difference in sampling depths that ranged from 0 to 0.2 m (Supplement Table S1) should be acceptable."

- Differences in land uses are only addressed in the section on land use change, but are very likely to be a main driver of the differences in the entire dataset. Phosphorus (and carbon) inputs widely differ among agricultural and natural systems, and this should be addressed (more prominently) in the current study.

Most studies considered in the meta-analysis report data on (semi-)natural soils as well as on cropland soil that are located in close vicinity. The minority of studies that only report data on (semi-)natural sites is distributed over the whole latitudinal range (latitudes: 3, 8, 43, and 57°). Thus, the land-use type does not bias the analyses of the effect of latitude. In order to show this more clearly, I added a column to Table 1, which indicates the land-use type of the study sites.

- The effect of soil pH is not addressed. The author mentions adsorption of (organic) P to metal (hydr)oxides as one of the main drivers. The affinity of different P compounds to adsorb to reactive surfaces strongly depends on pH, as the author mentions in the introduction. This point does not receive any further attention in the rest of the paper.

The reason why the effect of pH is not further addressed here is that none of the primary studies reports the pH of the different particle size fractions. Moreover, in order to conduct an analysis on how pH affects the affinity of different P compounds to sorb to reactive surfaces one would need more precise data on the nature of the P compounds and on the reactive surfaces in each particle size fraction, which is not provided in the studies.

Considering that these factors remain largely unaddressed and taking into account the limited size of the analysis (only 11 studies were incorporated), I think the manuscript is unfit for publication in its current form. Additionally, the manuscript is not written very elegantly. There are many repetitions, errors, and grammatically incorrect sentences throughout the manuscript that give it an unfinished look.

Specific comments

Introduction:

- The introduction can be written shorter and more concisely towards the goal of the study. Some examples:

- The first two paragraphs (p. 1, l. 5) can be summarized in two or three short sentences that indicate P is important for organisms and distinguishing between IP and OP in soil.

- The information on different inositol phosphates (p. 1, I. 18) is not relevant here.

- The entire paragraph on sorption of P monoesters and diesters might be replaced with one or two sentences on the difference in sorption affinity between higher-order inositol phosphates, P diesters and inorganic P (orthophosphate). The last remark on competition between inorganic P and organic carbon comes out of the blue and does not serve any purpose here.

- The section on the C:P ratio of the plant input (p. 2, l. 11) can be shortened. There is no need to discuss the results of these meta-analyses in detail. Additionally, in I. 14 the author suggests a decreasing order, but the signs suggest an increasing order.

I shortened some parts of the Introduction according to the reviewer's recommendations.

- On p. 2, I. 22 the author describes the objectives of the study, but the effect of soil depth and land use are not mentioned anywhere in the introduction.

The idea behind integrating data on land-use change from (semi-)natural vegetation to cropland is to test how this disturbance affects the distribution of OP, OC and IP among particle size fractions with the aim to learn about the sequestration of OP, OC and IP in the three different particle size fractions. I added some lines about how OP in soil depends on land-use and soil depth in the Introduction.

- When the effect of P inputs on OC/OP ratio is discussed (p. 3, l. 11), the author mentions plant material, but not other inputs like (organic or mineral) P fertilizer or animal waste/manure. This would be acceptable if the meta-analysis was restricted to forest ecosystems, but that is not the case.

I added the words "anthropogenic inputs such as fertilizer and manure".

Materials and Methods:

- The search method is not reproducible. I would expect a search query.

I added the lines "via Google Scholar and ScienceDirect, using the terms "particle size fractions", "soil organic phosphorus", "organic matter fractionation", "soil phosphorus", and "organic matter" in all possible combination".

- It is not clear to me why studies that report only two pools of organic P in the Hedley fractionation are included?

This seems to be a misunderstanding. All studies had to report OP concentrations of the three particle size fractions, namely sand, silt, and clay. However, studies that calculated organic P in the particle size fractions based on two pools of organic P determined by the Hedley fractionation, were included in the meta-analysis.

- Saunders and Williams describe several methods in their paper. I assume the author is referring to the ignition method.

**Yes, I added this information.**

- In the Saunders and Williams ignition method, it is organic P that is determined as the difference between total and inorganic P, not the other way around as the author states on page 4. *I corrected this.*

- Both methods to determine the soil P fractions have their pros and cons, yet the influence of the procedure or the possible implications of including different methods are not discussed. I would expect a section on this topic or an analysis to the effect of the method used in the studies, as it can significantly affect the results and interpretation.

*I inserted a section entitled "Methodological Considerations" at the beginning of the Introduction, in which I address this topic.*

- Not enough information is provided on the NMR analysis. NMR analysis results may depend on many variables, including the type of NMR (solid vs. liquid state), the extraction procedure used, delay and acquisition times, etc.

*I added the sentence "All studies that reported NMR data extracted the soil in diluted NaOH solution, centrifuged and dialyzed the extracts, and finally measured 31P-NMR spectra of the liquid samples on a NMR spectrometer (Bruker Instruments)."*

**Results and Discussion:**

- In general, there is not enough comparison to the existing literature on the implications of the findings.

I added more comparisons to previous studies and included the following references; Campbell et al., 1986; Christensen et al., 1992; Cross and Schlesinger, 1995; Gérard, 2016; Motavalli and Miles, 2002; Turner et al., 2002; Negassa and Leinweber, 2009; Yang and Post, 2011; Menezes-Blackburn et al., 2018

- The soil clay fraction usually contains more Fe and Al oxides as mentioned in the discussion. It is a shame there is no information available on the Fe and Al content of the soils in the studies, as this could have helped strengthen this point. Adsorption to clay should not be overlooked here either, as clay minerals can substantially contribute to the reactive surface area of a soil (see Gerard et al. 2016 Geoderma 262:213-226)

I agree with the reviewer. I added that adsorption of P to clay also plays an important in the stabilization of OP compounds, and I added the study by Gérad (2016) as a reference.

- The conclusion about increased P losses from erosion (p. 8, l. 17) looks a little odd, without any context. The fact that the soil clay fraction is richer in organic material and nutrients is well known, so this conclusion is of limited novelty.

I agree, and I removed the sentence.

- I am not convinced that the lower OC:OP ratio in the clay size points towards a further degraded OM source (P. 8, I. 20). It might indicate that it contains OM of a different origin, or a preferential adsorption of organic compounds with a relatively high P content.

I agree. I reworded the section in the following way: "The low OC:OP ratio of the clay size fraction likely results from a preferential adsorption of P-rich organic compounds to AI and Fe oxides and clay minerals in the clay size fraction (Gérad, 2016). The strong sorption of P to mineral surfaces likely stabilizes P-containing OM against decomposition, which might lead to an enrichment of P-rich compounds. The sportive stabilization of P-rich OM might contribute to the relatively high age of the OM in the clay size fraction compared to the two other particle size fractions"...

- The ratio between P monoesters and diesters seems very low (relatively high abun- dance of diesters) compared to other previous analyses (e.g. Menezes-Blackburn et

al. 2018 Plant & Soil 427:5-16). I am wondering why this is the case.

The mentioned study by Menezes-Blackburn et al. (2018) summarizes data obtained using a large number of different approaches. In contrast, in my study, the NMR data has all been obtained applying the same method. Therefore, it is not very surprising that the results of the two meta-analyses differ. Different sample preparation for NMR analysis leads to different results, as explained by anonymous referee#1. This problem does not occur in my meta-analysis since all NMR data considered have been obtained using the same technique.

- It seems to me that latitude and MAT should be strongly correlated. Yet their relation to the measured P pools is dissimilar. This topic should be disentangled and explained further.

The reason for this is that some studies have been conducted on high altitudinal sites (Solomon and Lehmann, 2000; Solomon et al., 2002; Makarov et al., 2002). I added some lines about the relationship between latitude and MAT to the Discussion. In addition, I removed the results of the multiple regressions that contained both latitude and MAT or latitude and MAP as independent variables.

- Land-use change had no effect on P stocks in this study. The data do thus not support the suggestion made in the last sentences on this section (p. 10, I. 21). As data on additional inputs (fertilizer, manure) and outputs (crop yield) of C and P are not provided, any conclusions on mineralization or leaching of organic P are purely speculative and should be avoided. *I removed the sentence*.

- In the conclusion, the author makes a remark about the potential of soil OP to replace P fertilizer inputs. This comes out of the blue and should be discussed in a discussion section first The last sentences of the conclusion links back to the first sentences of the introduction. However, I understand that it might come a little out of the blue. Therefore, I discuss this point already before in the Discussion section, in the revised version of the manuscript.

**Refences**

Campbell, C. A., Biederbeck, V. O., Selles, F., Schnitzer, M., & Stewart, J. W. B. (1986). Effect of manure and P fertilizer on properties of a Black Chernozem in southern Saskatchewan. Canadian journal of soil science, 66(4), 601-614.

Christensen, B. T. (1992). Physical fractionation of soil and organic matter in primary particle size and density separates. In Advances in soil science (pp. 1-90). Springer, New York, NY.

Cross, A. F., & Schlesinger, W. H. (1995). A literature review and evaluation of the. Hedley fractionation: Applications to the biogeochemical cycle of soil phosphorus in natural ecosystems. Geoderma, 64(3-4), 197-214.

Darch, T., Blackwell, M. S., Hawkins, J. M. B., Haygarth, P. M., and Chadwick, D.: A meta-analysis of organic and inorganic phosphorus in organic fertilizers, soils, and water: Implications for water quality. Crit Rev Environ Sci Technol, 44(19), 2172-2202, 2014.

Gérard, F. (2016). Clay minerals, iron/aluminum oxides, and their contribution to phosphate sorption in soils—A myth revisited. Geoderma, 262, 213-226.

Menezes-Blackburn, D., Giles, C., Darch, T., George, T. S., Blackwell, M., Stutter, M., ... & Brown, L. (2018). Opportunities for mobilizing recalcitrant phosphorus from agricultural soils: a review. Plant and Soil, 427(1-2), 5-16.

Motavalli, P., & Miles, R. (2002). Soil phosphorus fractions after 111 years of animal manure and fertilizer applications. Biology and Fertility of Soils, 36(1), 35-42.

Negassa, W., & Leinweber, P. (2009). How does the Hedley sequential phosphorus fractionation reflect impacts of land use and management on soil phosphorus: a review. Journal of Plant Nutrition and Soil Science, 172(3), 305-325.

*Turner, B. L., Papházy, M. J., Haygarth, P. M., & McKelvie, I. D. (2002). Inositol phosphates in the environment. Philosophical Transactions of the Royal Society of London B: Biological Sciences, 357(1420), 449-469.*

Yang, X., & Post, W. M. (2011). Phosphorus transformations as a function of pedogenesis: A synthesis of soil phosphorus data using Hedley fractionation method. Biogeosciences, 8(10), 2907-2916.